# Measurement of Chromosomal Arms and FISH Reveal Complex Genome Architecture and Standardized Karyotype of Model Fish, Genus *Carassius*

**DOI:** 10.3390/cells10092343

**Published:** 2021-09-07

**Authors:** Martin Knytl, Nicola Reinaldo Fornaini

**Affiliations:** Department of Cell Biology, Faculty of Science, Charles University, 12843 Prague, Czech Republic; fornainni@natur.cuni.cz

**Keywords:** chromosome, karyogram, in situ hybridization, *i* value, *q*/*p* arm ratio, *Carassius auratus*, *Carassius carassius*, *Carassius gibelio*

## Abstract

The widely distributed ray-finned fish genus *Carassius* is very well known due to its unique biological characteristics such as polyploidy, clonality, and/or interspecies hybridization. These biological characteristics have enabled *Carassius* species to be successfully widespread over relatively short period of evolutionary time. Therefore, this fish model deserves to be the center of attention in the research field. Some studies have already described the *Carassius* karyotype, but results are inconsistent in the number of morphological categories for individual chromosomes. We investigated three focal species: *Carassius auratus*, *C. carassius* and *C. gibelio* with the aim to describe their standardized diploid karyotypes, and to study their evolutionary relationships using cytogenetic tools. We measured length (q+p
length) of each chromosome and calculated centromeric index (*i* value). We found: (i) The relationship between q+p
length and *i* value showed higher similarity of *C. auratus* and *C. carassius*. (ii) The variability of *i* value within each chromosome expressed by means of the first quartile ($Q_{1}$) up to the third quartile ($Q_{3}$) showed higher similarity of *C. carassius* and *C. gibelio*. (iii) The fluorescent in situ hybridization (FISH) analysis revealed higher similarity of *C. auratus* and *C. gibelio*. (iv) Standardized karyotype formula described using median value ($Q_{2}$) showed differentiation among all investigated species: *C. auratus* had 24 metacentric (*m*), 40 submetacentric (sm), 2 subtelocentric (st), 2 acrocentric (*a*) and 32 telocentric (*T*) chromosomes (24m+40sm+2st+2a+32T); *C. carassius*: 16m+34sm+8st+42T; and *C. gibelio*: 16m+22sm+10st+2a+50T. (v) We developed R scripts applicable for the description of standardized karyotype for any other species. The diverse results indicated unprecedented complex genomic and chromosomal architecture in the genus *Carassius* probably influenced by its unique biological characteristics which make the study of evolutionary relationships more difficult than it has been originally postulated.

## 1. Introduction

Karyotype analysis is a fundamental approach by which chromosomes are arranged into homologous pairs with a respect to certain morphological categories. In the broader sense, the term karyotype is also referred to as a complete number of chromosomes describing typical taxon, species, biotype or individual [1]. Homologous chromosome pairs and morphological categories are determined based on the ratio of the long (*q*) and short (*p*) arm and the position of the centromere (centromeric index, *i* value). If the centromere is situated in median, submedian, subterminal or terminal region of the chromosome, the morphological category might be designated as metacentric (*M*, *m*), submetacentric (sm), subtelocentric (st) or acrocentric (*a*)/telocentric (t,T), respectively [2]. Two edge points that are located on median and terminal position sensu stricto have centromeric index 50 (assigned as *M*) and 0 (assigned as *T* chromosomes), respectively [2]. In order to maximize the diagnostic information obtainable from a chromosome preparation, the precise arrangement of individual chromosomes within a single image might be used as a standardized format. The term karyogram can be also used for a graphical depiction of chromosome complements [3,4].

Knowledge of the karyotype is necessary in (cyto)genetics, or related fields, in order to study chromosomal rearrangements and abnormalities [5,6], the identification of sex chromosomes [7,8], and/or chromosome-specific genes [9]. Although chromosomal changes, (i.e., chromosomal losses, duplications, rearrangements), are strictly linked to a certain locus, without the knowledge of detailed karyotype, it is not possible to precisely identify chromosomal aberration, mutation or syndrome [10,11]. Heteromorphic sex chromosomes (morphologically distinct in male or female individuals) can be distinguished by classic cytogenetic procedures based on Giemsa-staining karyogram of both sexes and/or by chromosome painting and fluorescent in situ hybridization (FISH) [12,13]. Cytogenetic mapping and localization of single-copy gene regions in the genome generally reveal both intra- and interchromosomal rearrangements (i.e., inversions, insertions, deletions or duplications) determined using karyograms [9]. Moreover, gene loci, which have not yet been mapped within a genome, can be assigned to a specific chromosome within a standardized karyogram [14].

In some model vertebrate species, a low/moderate number of chromosomes, meaning up to approximately 50 chromosomes, of relatively large-size allows for easy identification of each chromosome according to q/p arm ratio and centromeric index [15]. Each chromosome is usually assigned a numeric code, necessary in evolutionary studies pertaining to patterns of orthologous chromosomes between species. Orthologous chromosomes are inferred to be descended from the same ancestral chromosome separated by a speciation event. This concept is also referred to as shared synteny. In a non-model organism with a high number of relatively small chromosomes, identification of chromosomes into categories are inconsistent (e.g., Knytl et al. [16] vs. Kobayasi et al. [17] vs. Ojima and Takai [18]), as is that case with cyprinid fish from the genus *Carassius*. Although a large number of studies describing *Carassius* karyotype, there is no study proposing a standardized karyogram based on exact measurements of the difference between *q* and *p* chromosome arm, arm ratio and centromeric index in any *Carassius* species.

The genus *Carassius* belongs to the monophyletic paleotetraploid tribe, Cyprinini sensu stricto [19], within the family Cyprinidae (ray-finned fishes, Teleostei). Several species have been described within the genus *Carassius* and three of them are widely used in cytogenetic research: (i) *Carassius carassius* (Crucian carp), native and threaten in many European countries [20,21], is diploid with chromosome number 2n=4x=100 [17], where *n* refers to the number of chromosomes in each gamete of extant species, and *x* refers to the number of chromosomes in a gamete of the most recent diploid ancestor of the extant species. Most members of the family Cyprinidae contain 25 chromosomes in each gamete, considered as the most recent diploid ancestral state of extant *Carassius* [22]. (ii) *Carassius auratus* (goldfish), well known for its colourful varieties, bizarre shapes of body and formation of domesticated and feral populations [23]. (iii) *Carassius gibelio* (Silver Prussian carp) has recently spread throughout most continental waters, as a result of relocation from native habitats most likely by humans [24,25,26,27]. In addition, *Carassius auratus* and *C. gibelio* form diploid (2n=4x=100), triploid (3n=6x≈150), and tetraploid biotypes (4n=8x≈200) [28,29,30].

Here we described standardized karyotype of diploid *Carassius* species with 100 chromosomes (*C. auratus*, *C. carassius* and *C. gibelio*) based on measuring of chromosomal q+p arm length, q/p arm ratio, and *i* value. We performed FISH experiments with ribosomal probes, and analysed values of q+p
length and *i* in order to characterize the inter-species differences between three *Carassius* species, which commonly co-occur in European waters (*C. carassius* and *C. gibelio*), and bred/distributed worldwide as part of the pet trade (*C. auratus*).

## 2. Materials and Methods

### 2.1. Fish Sampling and Origin

*Carassius auratus* were obtained via the aquarium trade (transported to Czech Republic from Israel). *Carassius carassius* was captured in the Elbe River basin closed to the city Lysá nad Labem, Czech Republic [16,31] and in small pond in Helsinki, Finland [32]. *Carassius gibelio* originated from the Dyje Rive basin, South Moravia, Czech Republic [28].

### 2.2. Chromosome Analysis

Both males and females of each species (*C. auratus*, *C. carassius*, *C. gibelio*) were used for karyotype analysis. In *C. auratus*, mitotic activity was stimulated by intraperitoneal injection of 0.1% CoCl${}_{2}$ [32]. Somatic chromosomal complements were synchronized in metaphase of cell mitotic division using 0.1% colchicine (Sigma, St. Louis, MO, USA). Chromosome suspension was generated from the cephalic kidney [33] and stored in fixative solution (methanol: acetic acid, 3:1) at −20 °C, before being observed on a glass slide and stained with 5% Giemsa solution in 1× PBS. The chromosome suspensions used in this study have been used in prior analyses of *C. carassius* and *C. gibelio* [16,28,31,32] meaning that chromosomal suspensions were stored 3–7 years in −20 °C. Chromosomal suspensions were spun in a centrifuge, and fresh methanol and acetic acid were added every three months to maintain the constant ratio of methanol and acetic acid in fixative solution.

### 2.3. Measurements of q+p Chromosomal Arm Length

Photos of mitotic metaphase were taken using a Leica DFC 7000T camera and Leica DM6 microscope equipped with a EL6000 (metal halide) fluorescence illumination. A total of ten high-quality photos were taken for each species, five male and five female metaphases, from which measurements of the length of *q* and *p* chromosomal arms were taken. The centromere of each chromosome was identified as the narrowest part of a chromosome (Appendix A). Both arms of each chromatid (i.e., long arm 1 (q1), long arm 2 (q2), short arm 1 (p1), short arm 2 (p2)) were measured using ImageJ (V 1.53i) [34]. Chromosomal length (length), difference between *q* and *p* arm (*d*), q/p arm ratio (*r*) and centromeric index (*i*) were calculated according following formulas adopted from Levan et al. [2]:(1)$$p=\frac{p1+p2}{2}\;q=\frac{q1+q2}{2}$$
(2)$$length=p+q\;d=q-p\;r=\frac{q}{p}\;i=\frac{100}{r+1}$$

Measurements of *p*, *q*, length, *d*, *r*, and *i* were calculated as values in pixels from each image/metaphase and were further analyzed using R software for statistical computing (V 4.1.0) [35] and RStudio environment (V 1.4.1717) [36]. The *i* value was selected as a crucial characteristics for karyotypic analysis because it generally ranges from 0 to 50 [2]. The *r* value generally ranges from 1 to *∞* and thus this value is not suitable for graphical expression of results. The length and *i* value were used for calculation of an arithmetic mean of length (mean_length) and an arithmetic mean of *i* (mean_i) separately for each *Carassius* species. The mean_length and mean_i of each species were plotted (plot function in R).

### 2.4. Standardized Carassius Karyotype

  Two highest *i* values (one chromosomal pair) were dissected from each of the metaphases (ten metaphases in total) and put into additional data frame as a numeric vector. This group of *i* values were identified as chromosome 1 (hereafter chr1). The third and fourth highest *i* values were dissected from each metaphase and identified as chr2 and so on up to two chromosomes with the lowest *i* values identified as chr50. This function(){} is named Select_chrome and shown in the supplemental results. All *i* values of each identified chromosome were plotted (boxplot function in R) separately in each species. The default interquartile range 1.5 was used for elimination of extreme values (errors). Output value ($stats) was called in R. The *i* values within the the minimum ($Q_{0}$), first quartile ($Q_{1}$), median ($Q_{2}$), third quartile ($Q_{3}$) and maximum ($Q_{4}$) were investigated in detail, especially if some *i* values of each individual chromosome were shared among focal *Carassius* species. Chromosomal categories (*M*, *m*, sm, st, *a* and *T*) were determined according $Q_{2}$ of *i* values in each *Carassius* species. A morphological category of each chromosome was determined according chromosomal nomenclature Levan et al. [2]. The following Table 1 shows boundaries between each morphological category:

### 2.5. Preparation of 5S and 28S Ribosomal Probes

New 5S PCR primers 5S_F (5${}^{\prime }$–CAGGGTGGTATGGCCGTAGG–3${}^{\prime }$) and 5S_R (5${}^{\prime }$–AGCGCCCGATCTCGTCTGAT–3${}^{\prime }$) were designed according to the 5S gene of the western clawed frog, *Xenopus tropicalis*. The *X. tropicalis* genomic sequence is available on Xenbase, accessed on 11 June 2020 (http://www.xenbase.org). The 28S primers 28S_A (5${}^{\prime }$–AAACTCTG GTGGAGGTCCGT–3${}^{\prime }$) and 28S_B (5${}^{\prime }$–CTTACCAAAAGTGGCCCACTA–3${}^{\prime }$) used in this study were designed by Naito et al. [37]. Total genomic DNA (gDNA) was extracted from pectoral fin tissue using the DNeasy Blood & Tissue Kit (Qiagen, Hilden, Germany) according to manufacturer’s instructions. *Carassius gibelio* and *X. tropicalis* gDNA were used as a template for 5S and 28S PCR amplification, respectively. Primers were made by Integrated DNA Technologies (Coralville, IA, USA). The PPP Master Mix (Top-Bio, Prague, Czech Republic) was used for efficient amplification of ribosomal gene fragments. The temperature profile for the non-labelling amplification of the 5S and 28S loci followed Top-Bio instructions: initial denaturation step for 1 min at 94 °C, followed by 35 cycles (94 °C for 15 s, 53 °C for 15 s and 72 °C for 40 s) and a final extension step at 72 °C for 7 min. The obtained PCR products were separated on a 1.25% agarose gel with TA buffer and extracted from the gel using MicroElute Gel Extraction Kit (Omega Bio-tek, Norcross, GA, USA) according to manufacturer’s instructions. The PCR amplicons were sequenced and mapped using blastn algorithm for finding out of the locus- and species-specificity of amplification. Subsequently, the 5S and 28S ribosomal DNA (rDNA) loci were indirectly labelled by Digoxigenin-11-dUTP (Roche, Mannheim, Germany) and Biotin-16-dUTP (Jena Bioscience, Jena, Germany), respectively, by PCR reaction again. Taq DNA polymerase (Top-Bio) was used for labelling instead of The PPP Master Mix which was used for non-labelling amplification. Conditions for labelling PCR of the 5S and 28S loci were adopted from Sember et al. [38] and slightly modified as follows: initial denaturation step for 3 min at 94 °C, followed by 30 cycles (94 °C for 30 s, 53 °C for 30 s and 72 °C for 40 s) with final extension step at 72 °C for 10 min. The PCR product was separated on an 1% agarose gel with TBE buffer and purified using E.Z.N.A. Cycle Pure Kit (Omega Bio-tek) according to manufacturer’s instructions.

### 2.6. Fluorescent In Situ Hybridization

Cell suspensions prepared from each *Carassius* species were spread onto clean microscopic slides, which were subsequently used for FISH on the same day. The 5S probe from *C. gibelio* was used for chromosomal spreads of *C. auratus*, *C. carassius* and *C. gibelio*. The 28S probe from *X. tropicalis* was used for *C. auratus*, *C. carassius* and *C. gibelio* chromosomal spreads. In total 44 μL of the hybridization mixture containing 100 ng of either 5S or 28S rDNA probe, 50% deionized formamide, 2× SSC, 10% dextran sulphate and water was placed on a slide and covered with a 24 × 50 mm coverslip. Both probe and chromosomal DNA were denatured in a PCR machine with special block for slides at 70 °C for 5 min [9]. Hybridization, post-hybridization washing, blocking reaction and visualization of 5S and 28S rDNA signals were carried out as described in Knytl et al. [39]. The Digoxigenin-11-dUTP/Biotin-16-dUTP labelled probe was detected by Anti-Digoxigenin-Fluorescein (Roche)/CY^TM^ 3-Streptavidin (Invitrogen, Camarillo, CA, USA), respectively, diluted according to manufacturer’s instructions. Chromosomes were counterstained with ProLong^TM^ Diamond Antifade Mountant with DAPI (Invitrogen by Thermo Fisher Scientific, Waltham, MA, USA). At least 20 metaphase spreads in total were analysed per individual.

## 3. Results

### 3.1. Karyotype Analysis

Number of chromosomes in all studied species (*C. auratus*, *C. carassius* and *C. gibelio*) was uniform (2n=4x=100) Appendix A. We did not find any differences in chromosomal morphology between male and female individuals. This finding confirmed homomorphic sex chromosomes at least in diploid biotypes of the genus *Carassius*.

### 3.2. Interspecies Variability Based on Chromosomal Length

Detailed description of R analysis including R scripts are given in Appendix A. All steps outlining how the measured values were calculated and processed into tables and plots are shown on the *C. auratus* dataset. The mean_i and mean_length of each individual chromosome were plotted onto *x* axis and *y* axis, respectively, (Figure 1). Morphological chromosomal categories *m*, sm, st, *a* and *T* are present in each species. Extreme category, such as *T*, is represented by 19, 26, and 25 chromosomes of *C. auratus*, *C. carassius* and *C. gibelio*, respectively (i.e., 19, 26, and 25 points lie on gray dashed vertical lines of interval 0). The *T* chromosomes have small mean_length, approaching 0. Other chromosomal categories, such as *M*, are not represented in any species—no *i* value reached 50, with the highest *i* of 48.778 found in chromosome 1 of *C. gibelio*. Chromosomal length of *C. auratus*, *C. carassius*, and *C. gibelio* range from 23.340–40.614, 26.799–44.092, and 15.205–29.496, respectively. It is evident that chromosomes of *C. gibelio* are generally smaller (mean of q+p
*length* was 21.36) than those of both *C. auratus* (31.53) and *C. carassius* (35.05). For the minimum of errors we calculated mean_length and mean_i from a group of ten measured q+p values, i.e., each dot on Figure 1 is arithmetic mean resulted from ten measured q+p values. Potential influences on chromosomal length are discussed (Section 4).

### 3.3. Interspecies Variability Based on Centromeric Index Linked to Individual Chromosomes of the Whole Chromosomal Complement, and Standardized Carassius Karyotype

The *i* values were assigned to each individual chromosome by dissection of chr1–50 from each metaphase (also Section 2.4, R protocol is described in the section of R analysis of Appendix A) and plotted onto *x* and *y* axis (Figure 2). Each box depicts a group of chromosomes dependent on *i* values ordered from most metacentric chromosomes on the left side of plots (chr1) to most telocentric chromosomes on the right side of plots (chr50). In general, each species has six to seven chromosomes that were highly variable in *i*, i.e., chr31–37 in *C. auratus*, chr27–33 in *C. carassius* and chr23–29 in *C. gibelio*. The $Q_{1}$ and $Q_{3}$ cover *i* values from 0 to 29.342 in *C. auratus*, from 0 to 23.9 in *C. carassius* and from 0 to 25.747 in *C. gibelio*. *Carassius auratus* has the highest variability of *i* value.

If some *i* values are shared among *C. auratus*, *C. carassius* and *C. gibelio*, the range of *i* values of each individual chromosome (chr1–50) from ten metaphases were compared with the range of *i* values of other two orthologs of each of other two species (ten metaphases from each species). The $Q_{1}$–$Q_{3}$ range of *i* value within chr1 of *C. auratus* was compared with the $Q_{1}$–$Q_{3}$ range of *i* value within chr1 of *C. carassius* and *C. gibelio*, similarly the $Q_{1}$–$Q_{3}$ range of *i* value within chr2 of *C. auratus* was compared with the $Q_{1}$–$Q_{3}$ range of *i* value of *C. carassius* and *C. gibelio* etc. up to the $Q_{1}$–$Q_{3}$ range of *i* value within chr50 of *C. auratus*, *C. carassius* and *C. gibelio*. As an additional analysis, multi-plot with 50 separate box plots (chr1–50) was generated, each box plot was composed of orthologous chromosomes of all three species (Figure 3).

Based on *i* value (Figure 1 and Figure 3), we concluded that:Each orthologous chromosome of *C. carassius* and *C. gibelio* shared *i* values within $Q_{1}$–$Q_{3}$ range and therefore we consider that karyotypes of *C. carassius* and *C. gibelio* to be more similar based on *i* value.Chromosomes 12–30 of *C. auratus* were represented by different *i* values within $Q_{1}$–$Q_{3}$ range those of *C. carassius* and *C. gibelio*. Therefore, we consider the karyotypes of *C. auratus* to be most variable for this parameter.Chromosomes 8, 11 and 31 of *C. auratus* were represented by different *i* values within $Q_{1}$–$Q_{3}$ range those of *C. gibelio* but *C. carassius* shared *i* values in these chromosomes with both *C. auratus* and *C. gibelio*.

The $Q_{2}$ range (median) of *i* value was used for determination of a standardized karyotype formula. Chromosome1 of *C. auratus*, *C. carassius* and *C. gibelio* with median of *i* value 47.87 (CAU_median_i column of Table 2), 48.28 (CCA_median_i) and 48.55 (CGI_median_i), respectively, were identified as *m* chromosomes in all three *Carassius* species (CAU_category, CCA_category, CGI_category columns of Table 2). Karyotypes all three *Carassius* species were different in the number of chromosomal categories sensu Levan et al. [2]. The number of chromosomes in categories found out by arithmetic mean (Figure 1) slightly differed from the number of chromosomes in categories determined by $Q_{2}$ range (Figure 2). The $Q_{2}$ range eliminated errors and thus we inferred standardized karyotype formula according to the $Q_{2}$ range.
(3)Carassiusauratus:24m+40sm+2st+2a+32T
(4)Carassiuscarassius:16m+34sm+8st+42T
(5)Carassiusgibelio:16m+22sm+10st+2a+50T

### 3.4. Fluorescent In Situ Hybridization with 5S and 28S Ribosomal Probes

The PCR amplification of 28S and 5S rDNA locus resulted consistently in approximately 300 bp long fragments. Searches, using the blastn algorithm, confirmed the locus- and species-specificity of each amplicon: 100% identity with 28S rRNA of *X. tropicalis* (accession numbers XR_004223792–XR_004223798), 95% identity with sequence of 5S rRNA of *C. gibelio* (accession number DQ659261). The amplified 5S rDNA sequence was deposited to the NCBI/GenBank database (accession number BankIt2492026 5s, MZ927820). Mapping of the 5S and 28S loci showed different patterns in the number and position within each investigated *Carassius* species (Figure 4). No differences between males and females were detected. The q+p arms of the FISH images were measured on chromosomes that bear positive rDNA loci because the FISH protocol involves denaturation step after which chromosomal structure is disrupted due to high temperature. The rDNA positive loci are mostly accumulated at the pericentromeric chromosomal region in *Carassius* and another cyprinid fishes [32,40,41] and thus the identification of the centromere and measurement of the q+p arms are precise on rDNA positive chromosomes. The *i* value of rDNA positive chromosomes (arrows on Figure 4) were assigned to the closest *i* value of the standardized karyotype from Table 2.

Summary of 5S rDNA (corresponding to non-nucleolar region). We found:Two more intensive (strong) 5S rDNA signals at the *p* arm of sm chromosomes (chr28, *i* value =30.06) and eight weak signals in *C. auratus*.Two strong 5S rDNA signals at the *p* arm of sm chromosomes (chr19, *i* value =30.60) and six weak signals in *C. carassius*.Two strong 5S rDNA signals at the *p* arm of sm chromosomes (chr17, *i* value =30.00) and eight weak signals in *C. gibelio*.

Summary of 28S rDNA (corresponding to nucleolar organizer region). We found:two strong 28S rDNA signals at the *p* arm of two sm chromosomes (chr16, *i* value =35.69) and two weak signals in *C. auratus*.two strong 28S rDNA signals at the *p* arm of two sm chromosomes (chr13, *i* value =34.72) and four weak signals in *C. carassius*.two strong 28S rDNA signals at the pericentromeric region of two *T* chromosomes (in the range of chr26–50, *i* value =0) and two weak 28S rDNA signals in *C. gibelio*.

## 4. Discussion

Diploid vs. polyploid biotypes—sexual vs. asexual reproduction—genetic vs. environmental sex determination: All these natural phenomena make *Carassius* a valuable experimental model for evolutionary studies. The study of these aforementioned unique characteristics of *Carassius* require basic cytogenetic techniques such as chromosome preparations and/or karyotype. Diploid biotypes of the genus *Carassius* have a karyotype consisting of 100 chromosomes (e.g., [28], this study), with some exceptions, i.e., 50, 94, 98, 102 or 104 chromosomes [22,42,43,44,45,46]. Karyotype formula of diploid *Carassius* is inconsistently defined by several authors (Table 3) and q+p arms have not been measured. An application of a technique of measurement, and the following determination of standardized karyotype is needed in order to clear up genome architecture and its evolution.

We generated a revised karyotype as a result of novel measurement and using statistical programming we described standardized karyotype of three species (*C. auratus*, *C. carassius* and *C. gibelio*). We measured the length of each individual chromosome of each species and identified each chromosome using *i* value. Groups of *i* values were divided into quartiles $Q_{0}$–$Q_{4}$. Our results based on median ($Q_{2}$) of *i* value for each chromosome revealed three different karyotypes; not one of these three karyotypes corresponded to any of the previously published karyotypes from Table 3. The karyotype of *C. auratus* had 24m, 40sm, 2st, 2a and 32T chromosomes (shortened formula 24m+40sm+36st − *T*). The karyotype of *C. carassius* had 16m, 34sm, 8st and 42T chromosomes (shortened formula 16m+34sm+50st − *T*). The karyotype of *C. gibelio* possessed 16m, 22sm, 10st, 2a and 50T chromosomes (shortened formula 16m+22sm+62st − *T*). These standardized karyotypes are likely highly reproducible and could be applied to the reconstruction of cytogenetic maps, comparative cytogenetics and genomics, or cytotaxonomy and karyosystematics. The designed R scripts (Appendix A) could be applied to the description of standardized karyotype for another species with relatively high number of chromosomes similar to that of *Carassius*.

In the wider range of *i* values ($Q_{1}$–$Q_{3}$) for each chromosome, we found karyotypes of *C. carassius* and *C. gibelio* more similar, as both *C. carassius* and *C. gibelio* shared some *i* values within $Q_{1}$–$Q_{3}$ range of corresponding orthologous counterparts. The *i* values in $Q_{1}$–$Q_{3}$ of chr8 and chr11–31 in *C. auratus* were significantly different those of *i* values in $Q_{1}$–$Q_{3}$ of *C. carassius* and *C. gibelio* (Figure 2 and Figure 3). This finding indicates higher divergence of *C. auratus* karyotype based on *i* value. The *i* value divergence might be supported by distinct origin of the samples used in this investigation—*C. auratus* used in this study was imported from Israel and was domesticated in ancient China [23]. Both *C. carassius* and *C. gibelio* used in this study originated from Europe and mostly from the Czech Republic (only four individuals of *C. carassius* were caught in Finland). A phylogenetic study of Asian–European *Carassius* suggests a closer evolutionary relationship exists between diploid *C. auratus* and diploid *C. gibelio* compared to *C. carassius* and diploid *C. auratus/gibelio* [66]. Thus, phylogenetic distance does not reflect *i* value divergence in *C. carassius*, as demonstrated in this study. The higher similarity between *C. carassius* and *C. gibelio*
*i* values could be caused by hybridization, as previously confirmed by molecular genetic and cytogenetic tools in several European *Carassius* populations [16,67,68]. Although, we can not exclude a hybrid origin for diploid *Carassius* individuals.

In addition, female meiotic drive might also affect *i* values [69]. Chromosomes of particular morphology might be preferentially transmitted to the egg during meiosis and the chromosomal complement of new offspring can be rapidly changed. Interestingly, polyploid *Carassius* biotypes have variable numbers of microchromosomes [16] which results in odd or variable chromosome numbers within polyploid *Carassius* [28]. The uniform chromosome numbers, we have described in diploid *Carassius*, are in accordance with expectations that female meiotic drive has never been shown in diploid *Carassius*.

We also showed a relationship between the mean of *i* value and the mean of q+p chromosomal length. Chromosomes of *C. gibelio* had smaller size (mean of chromosomal length was 21.36) those of *C. auratus* (31.53) and *C. carassius* (35.05, Figure 1). This difference in chromosomal length in *Carassius* might be promoted by altered chromatin spiralisation and condensation during cell cycle, especially during interphase and mitosis [70,71,72]. In this study, chromosomes were synchronized in metaphase by colchicine but some chromosomes can be fixed in prometaphase, early metaphase or late metaphase and we assume that slight differences in chromosomal length can be present in different *Carassius* species. The length of chromosomes might theoretically be influenced by storage time if the ratio or concentration of acetic acid and methanol is slightly changed [72]. To avoid these inaccuracies and preserving the chromosome suspension, we used a fixative solution with an acetic acid:methanol ratio of 1:3. This ratio was restored through the addition of fresh acetic acid and methanol. In addition, we statistically processed metaphases with different storage times, for which we found no difference in chromosome length. For the case of standardized karyotype, all previous negative effects that influence q+p arm length can be discounted because even if the length of chromosome had changed, the position of the centromere remains identical, thus *i* values never change [2]. In addition, the *i* value errors were eliminated through analysis of the median $Q_{2}$ range.

The correct order of chromosomes in *Carassius* karyograms remains difficult to determine. For our purposes, we arranged chromosomes according to decreasing *i* values. As such, chr1 has the highest *i* value and chr50 has the lowest *i* value. Using this approach describes a shared synteny of *i* value among *C. auratus*, *C. carassius* and *C. gibelio* (Figure 3). This method of numbering of chromosomes according *i* value in *Carassius* can be modified in future studies such as the numbering in human decreasingly according to q+p chromosomal length [73].

FISH analysis revealed two large non-nucleolar 5S loci and two large nucleolar 28S ribosomal loci in each of the *Carassius* species that were investigated; this indicates functional diploidy in evolutionary tetraploid biotype. Two strong rDNA signals were found in other studies which described *Carassius* karyotype [32,40,74]. The number of 5S rDNA loci (strong and weak signals in total) ranged from eight to ten, and 28S rDNA loci ranged from four to six, with a single chromosome-bearing rDNA locus differing in each *Carassius* species. We found higher similarity in a number of rDNA loci in *C. auratus* and *C. gibelio*, i.e., ten 5S and four 28S rDNA signals in these two species. *Carassius carassius* possessed eight 5s and six 28S rDNA signals. Knytl et al. [32] identified 18 5S and four 28S rDNA loci in diploid *C. carassius* originating from Finland waters. Spoz et al. [40] pointed out ten 5S and four 28S rDNA loci in *C. carassius* from Poland. Chinese *C. auratus* has 2–8 5S rDNA signals [74]. This study brought the first report of the FISH analysis used on diploid European *C. gibelio*, which belongs to phylogentically very diverse taxonomic group [66]. Our results showed variability in the number and position of ribosomal tandem repeats in *Carassius* which is in accordance with other rDNA studies [32,40,74]. This variability in rDNA loci has been shown between species of the same genus [38] and even between individuals of the same species for 18S and 28S loci [75,76]. The present variability of rDNA loci can be attributed to degree in heterochromatin condensation and nucleolar activity during mitosis [77].

Overall, we have described a standardized karyotype of three species of *Carassius* genus: *C. auratus*, *C. carassius* and *C. gibelio*. We compared these three species using cytogenetic tools: The arithmetic mean of the length of q+p chromosomal arm showed higher similarity of *C. auratus* and *C. carassius*, and a difference of *C. gibelio* (Figure 1). Analysis of median of the *i* value showed higher similarity of *C. carassius* and *C. gibelio*, and higher difference of *C. auratus* (Figure 2 and Figure 3). FISH confirmed a higher similarity of *C. auratus* to *C. gibelio*, compared to *C. carassius* (Figure 4). Thus, we can conclude that the genus *Carassius* has a very complex cytogenetic background, which can be distinguished using cytogenetic tools, with some inconsistencies in measuring propinquity and congeniality. The findings presented here are consistent with the presence of great genome variability and plasticity. Such geneome variability could be attributed to the occurance of rare natural phenomena such as interspecies hybridization, polyploidization, and/or alternation of reproduction modes within the genus *Carassius* [16,43,78,79].

## Figures and Tables

**Figure 1 cells-10-02343-f001:**
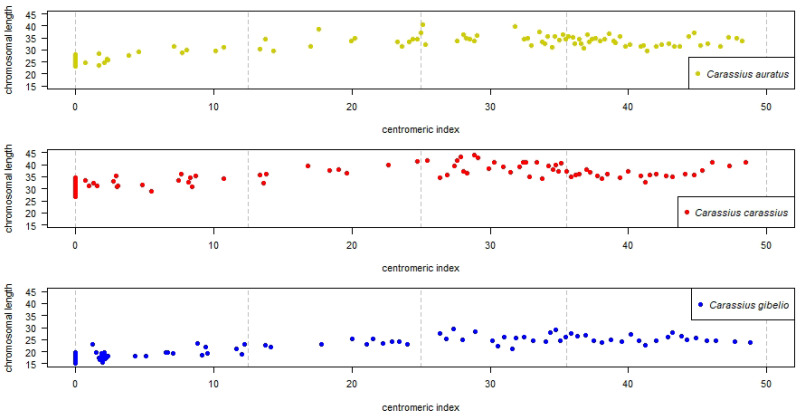
Relationship between centromeric index (*i*) and chromosomal length (length). Top plot shows chromosomes of *Carassius auratus* in yellow, middle plot shows chromosomes of *C. carassius* in red and bottom plot shows chromosomes of *C. gibelio* in blue. Chromosomal categories are bounded by gray dashed vertical lines which define intervals 0–12.5, 12.5–25, 25–37.5 and 37.5–50 corresponding to acrocentric (*a*), subtelocentric (st), submetacentric (sm) and metacentric (*m*) chromosomes, respectively. Both plotted *i* value and length are presented as an arithmetic mean of each chromosome.

**Figure 2 cells-10-02343-f002:**
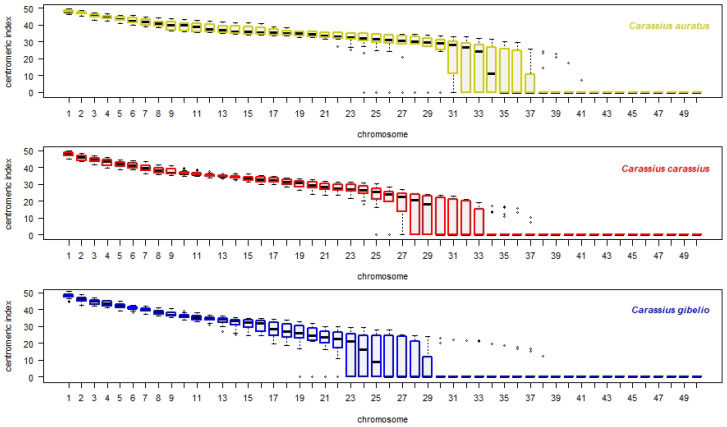
Intrachromosomal variability of *i* displayed on a whole chromosomal complement. Top plot shows haploid chromosomal complement (50 chromosomes) of *C. auratus* in yellow, middle plot shows 50 chromosomes of *C. carassius* in red and bottom plot shows 50 chromosomes of *C. gibelio* in blue. Each chromosome is linked to *i* value (*y* axis). Upper and lower whiskers show extreme values, the minimum ($Q_{0}$) and maximum ($Q_{4}$), respectively, boxes involve the first quartile ($Q_{1}$) and the third quartile ($Q_{3}$) group of values. Black line within box indicates median value of the dataset ($Q_{2}$). Outliers (errors) are drawn as black points.

**Figure 3 cells-10-02343-f003:**
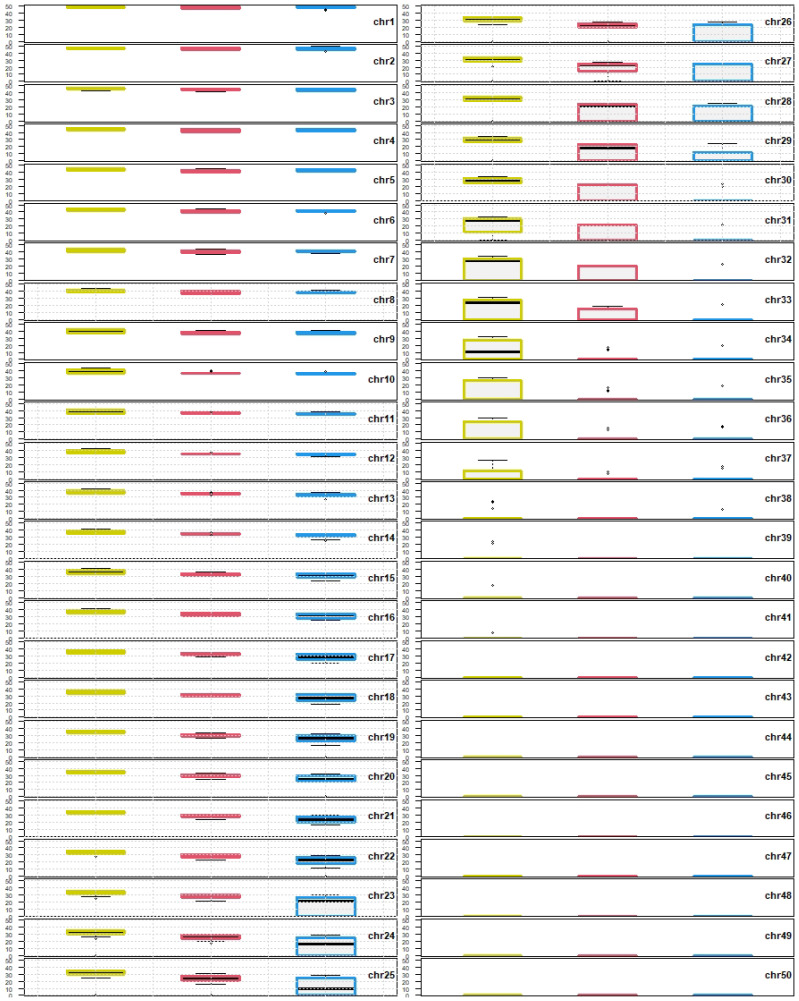
Box plots composed of orthologous chromosomes of each species. Chromosomes are ordered decreasingly according to the *i* value. Chromosomes on the top left of the figure are the most metacentric, chromosome 1 (chr1), chromosomes on the bottom right of the figure are the most telocentric (chr50). Yellow, red and blue boxes represents *C. auratus*, *C. carassius* and *C. gibelio*, respectively. Each chromosome is linked to *i* value (*y* axis). Upper ($Q_{0}$) and lower ($Q_{4}$) whiskers show extreme values, boxes involve $Q_{1}$ and $Q_{3}$ group of values. The black line within boxes indicates median value of the dataset ($Q_{2}$). Outliers (errors) are drawn as black points. The grid in some box plots represents significantly different *i* values within the $Q_{1}$–$Q_{3}$ range. Box plots without the grid share some *i* values within the $Q_{1}$–$Q_{3}$ range.

**Figure 4 cells-10-02343-f004:**
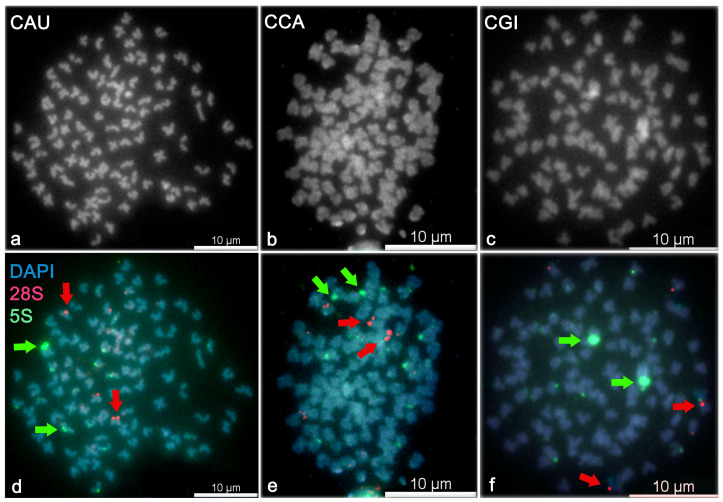
Double-colour fluorescent in situ hybridization with 5S and 28S ribosomal probes. DAPI-counterstained metaphase spreads show 100 chromosomes (B&W) in (**a**) *C. auratus* (CAU), (**b**) *C. carassius* (CCA) and (**c**) *C. gibelio* (CGI). The 5S (green) probe shows two more intensive (strong) signals and eight, six and eight less intensive (weak) signals in (**d**) *C. auratus*, (**e**) *C. carassius* and (**f**) *C. gibelio*, respectively. The 28S rDNA probe (red) reveals two strong signals and two, four and two weak signals in (**d**) *C. auratus*, (**e**) *C. carassius* and (**f**) *C. gibelio*, respectively. Green and red arrows correspond to 5S and 28S ribosomal loci, respectively.

**Table 1 cells-10-02343-t001:** Chromosomal nomenclature used for determination of chromosomal categories according to Levan et al. [2].

Centromeric Position	Arm Ratio	Centromeric Index	Chromosome Category
median sensu stricto	1.00	50	M (metacentric sensu stricto)
median	1.01–1.70	49.9–37.51	m (metacentric)
submedian	1.71–3.00	37.50–25.01	sm (submetacentric)
subterminal	3.01–7.00	25.00–12.51	st (subtelocentric)
terminal	>7.01	12.50–0.01	a/t (acro-/telocentric)
terminal sensu stricto	∞	0	T (telocentric sensu stricto)

**Table 2 cells-10-02343-t002:** Table of values used for determination of standardized karyotype. Each median ($Q_{2}$) of *i* values corresponds to certain chromosomal category of chr1–50. The medians of *i* value are sorted decreasingly. CAU = *C. auratus*, CCA = *C. carassius*, CGI = *C. gibelio*, m = metacentric, sm = submetacentric, st = subtelocentric, a = acrocentric, T = telocentric sensu stricto.

	CAU_median_i	CCA_median_i	CGI_median_i	CAU_category	CCA_category	CGI_category
chromosome1	47.87	48.28	48.55	m	m	m
chromosome2	47.07	45.68	46.12	m	m	m
chromosome3	45.74	44.74	44.99	m	m	m
chromosome4	44.81	43.52	43.47	m	m	m
chromosome5	43.46	42.11	42.53	m	m	m
chromosome6	42.43	41.08	40.89	m	m	m
chromosome7	41.63	39.58	39.89	m	m	m
chromosome8	40.84	37.90	38.42	m	m	m
chromosome9	40.13	36.81	36.91	m	sm	sm
chromosome10	39.67	36.17	35.84	m	sm	sm
chromosome11	38.75	35.79	35.41	m	sm	sm
chromosome12	37.65	35.19	34.73	m	sm	sm
chromosome13	36.77	34.86	34.25	sm	sm	sm
chromosome14	36.19	34.36	33.36	sm	sm	sm
chromosome15	35.88	33.66	32.61	sm	sm	sm
chromosome16	35.71	32.70	31.90	sm	sm	sm
chromosome17	35.48	32.22	28.62	sm	sm	sm
chromosome18	35.21	31.23	26.86	sm	sm	sm
chromosome19	34.87	30.54	26.07	sm	sm	sm
chromosome20	34.33	29.11	24.42	sm	sm	st
chromosome21	33.45	28.54	23.45	sm	sm	st
chromosome22	33.03	27.41	22.83	sm	sm	st
chromosome23	32.78	26.86	21.37	sm	sm	st
chromosome24	31.95	26.25	16.52	sm	sm	st
chromosome25	31.41	25.22	9.19	sm	sm	a
chromosome26	31.08	24.00	0.00	sm	st	T
chromosome27	30.70	22.57	0.00	sm	st	T
chromosome28	30.11	20.62	0.00	sm	st	T
chromosome29	29.75	18.13	0.00	sm	st	T
chromosome30	28.96	0.00	0.00	sm	T	T
chromosome31	28.17	0.00	0.00	sm	T	T
chromosome32	26.87	0.00	0.00	sm	T	T
chromosome33	24.41	0.00	0.00	st	T	T
chromosome34	11.19	0.00	0.00	a	T	T
chromosome35	0.00	0.00	0.00	T	T	T
chromosome36	0.00	0.00	0.00	T	T	T
chromosome37	0.00	0.00	0.00	T	T	T
chromosome38	0.00	0.00	0.00	T	T	T
chromosome39	0.00	0.00	0.00	T	T	T
chromosome40	0.00	0.00	0.00	T	T	T
chromosome41	0.00	0.00	0.00	T	T	T
chromosome42	0.00	0.00	0.00	T	T	T
chromosome43	0.00	0.00	0.00	T	T	T
chromosome44	0.00	0.00	0.00	T	T	T
chromosome45	0.00	0.00	0.00	T	T	T
chromosome46	0.00	0.00	0.00	T	T	T
chromosome47	0.00	0.00	0.00	T	T	T
chromosome48	0.00	0.00	0.00	T	T	T
chromosome49	0.00	0.00	0.00	T	T	T
chromosome50	0.00	0.00	0.00	T	T	T

**Table 3 cells-10-02343-t003:** Previously published karyotypes of diploid *C. auratus*, *C. carassius* and *C. gibelio* including information about sex and locality of the investigated individuals. NA = information not available, F = female, M = male.

Karyotype	Sex	Locality	References
***C. auratus***			
2n=94	NA	Japan	[42,47]
2n=96–104	F, M	NA	[48]
2n=100(12m+36sm+52st–a)	F, M	Japan	[49,50]
2n=104(46m+16sm+42a)	F, M	NA	[22]
2n=104(20m+72sm–st+12a)	NA	NA	[44]
2n=100(20m+40sm+40a)	F, M	NA	[17]
2n=100(16m+84sm–a)	NA	NA	[51]
2n=100(12m+36sm+52st–a)	F, M	China	[18,52,53]
2n=100(22m+30sm+48st–a)	F, M	China	[54,55,56]
***C. carassius***			
2n=104(20m+72sm–st+12a)	NA	NA	[44]
2n=100(20m+40sm+40a)	F, M	Netherlands	[17,57,58,59]
2n=100(20m+44sm+36a)	NA	France	[60]
2n=100	F, M	Bosnia	[61]
2n=50(20m+12sm+18st–a)	NA	Romania	[45]
2n=100	F, M	Czech Republic	[62]
2n=100(20m+36sm+44st–a)	F, M	Czech Republic	[16,31]
2n=100(20m+36sm+44st–a)	F, M	Poland	[40]
2n=100(20m+36sm+44st–a)	M	Finland	[32]
***C. gibelio***			
2n=94	F, M	Belarus	[43]
2n=100(20m+40sm+40a)	NA	River Amur	[63]
2n=98(48m–st+50a)	NA	Romania	[45]
2n=102(24m+36sm–st+42a)	F	Yugoslavia	[46]
2n=104(24m+36sm–st+44a)	M	Yugoslavia	[46]
2n=100(14m+24sm+62st–a)	F, M	Poland	[64]
2n=100(26m+38sm+36st–a)	F, M	Poland	[65]

## Data Availability

Sanger sequencing data are available on-line at the NCBI database, accessed on 26 August 2021 (https://www.ncbi.nlm.nih.gov/). All data generated by R Studio are not presented in this study and they may be available on request from the corresponding author. All steps generating data frames and plots in R were summarized on-line on GitHub, accessed on 26 August 2021 (https://www.github.com/) available on request. Basic q+p arm measurements can be provided in the form of RData file as R workspace, also upon request.

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
