# Peer review of "Measurement of Chromosomal Arms and FISH Reveal Complex Genome Architecture and Standardized Karyotype of Model Fish, Genus Carassius"

_cells, 2021, doi:10.3390/cells10092343_

Round 1

Reviewer 1 Report

Dear authors, 

I am a very enthusiastic of karyomorphometry and a think that the data from such analysis can provide valuable information, mainly for comparative karyotype study. 

I really appreciate the opportunity to read the paper, but I feel that a can not recommend the publication in the present form. As well, I do consider that the venue as Cells can not exclude good contribution only because there is no novelty or fancy methodologies applied, and conserving my ethical principles I would recommend major revisions.  

All my questions and concerns are highlighted in the pdf file by using Acrobat comments. 

I would like to emphasize: 

1 - Please provide the R scripts as supplementary material. It is fundamental for reproducibility. 

2 - Please cite Levan et al. 1964 in Table 2 and in all mentions in karyomorphometric data and etc.

3 - Were are the metaphases and karyotypes images? How the authors could identify the centromere? By checking the fish metaphases and the number of chromosomes. How precise are the measeruments?  I am very impressed with the ability to measure such chromosomes. I think that is possible that much error are being admitted by using such approach and this needs to be clarified by the authors.

4 - "Standardized karyotype": Please provide a carefully description that how this was achieved because it is not clear to me and the data provided do not allow checks. I thing that solely, karyomorphometrics do not precisely can be used considering the size and number of the chromosomes. And how about differences in chromosome condensation? How this could be overcome? 

Other comments are included in the file. 

Hope see the next version. 

All the best.

Reviewer 2 Report

This manuscript “Measurement of the Length of Chromosomal Arms and the FISH Approach Revealed Complex Genome Architecture and Standardized Katyotype of Fish Genus Carassius” presents the first detailed cytogenetic analysis of the genus Carassius. The biological characteristics of the genus, which have been listed and discussed, such as polyploidy, hybridization between species, clonality, could indicate the separation of species in the recent evolutionary time. Additionally, karyotype evolution plays an important role in divergent adaptation and speciation, so in this case the listed biological characteristics could reflect the wide species distribution. Results of this study indicate that different cytogenetic parameters show different levels of similarity among the analyzed species. Nevertheless, the obtained standardized karyotypes based on detailed measurements represent the most valuable part of this manuscript. The finding of functional diploidy in evolutionary tetraploid biotype, based on FISH is also very interesting.

Chromosome morphology is thought to be under female meiotic drive (King, 1993). This means that chromosomes of particular morphology are preferentially transmitted to the egg. It will be interesting to discuss this idea in light of the biology of studied species.

My biggest concern is the possibility of valid measurements on chromosomes of this size and this number in the karyotype. The first problem is the degree of chromosome condensation, which can influence the size and further chromosome variability, but I consider that the preparation of chromosomes was uniform and that the measurement was done in the same way. Hence, the comparison performed is probably the best you can get using standard cytogenetics methodology. Did you take care to cover all analyzed sites by measuring chromosome pairs at a total of 10 metaphases?

Some minor comments:

Line 10-11 “The dependence of q + p length and i value showed a higher similarity of C. auratus and C. carassius.” I do not think that “dependence” is the best word in this sentence. I would suggest changing it to “relationship between”.

 Lines 226-229 Chromosomal morphology is defined by length (q+p) and centromere position (i). Based on higher similarity in i you can’t concludes that C. carassius and C. gibelio are more closely related based on chromosomal morphology, bearing in mind that species differ significantly in the average length of chromosomes. They similarity based on i only.

Lines 230-232 in the average divergence of the chromosomal index (i) is dependent on the number of telocentric chromosomes mainly. The fact that C. auratus have the lowest number of telocentrics, makes the karyotype of this species the most variable for this parameter.

Taking into account altogether, from the concept and analysis to the presentation and discussion of the obtained results, I warmly recommend the manuscript to be accepted for publication in the Cells, with minor proposed changes.

Round 2

Reviewer 1 Report

All comments has been followed by the authors and I do appreciate to see published soon.